

# A Profile Hidden Markov Model to investigate the distribution and frequency of LanB-encoding lantibiotic modification genes in the human oral and gut microbiome

Calum J. Walsh[1,2], Caitriona M. Guinane[1], Paul W. O' Toole[2,3] and Paul D. Cotter[1,3]

[1] Teagasc Food Research Centre, Moorepark, Co. Cork, Ireland
[2] School of Microbiology, University College Cork, Co. Cork, Ireland
[3] APC Microbiome Institute, University College Cork, Co. Cork, Ireland

## ABSTRACT

**Background**. The human microbiota plays a key role in health and disease, and bacteriocins, which are small, bacterially produced, antimicrobial peptides, are likely to have an important function in the stability and dynamics of this community. Here we examined the density and distribution of the subclass I lantibiotic modification protein, LanB, in human oral and stool microbiome datasets using a specially constructed profile Hidden Markov Model (HMM).

**Methods**. The model was validated by correctly identifying known lanB genes in the genomes of known bacteriocin producers more effectively than other methods, while being sensitive enough to differentiate between different subclasses of lantibiotic modification proteins. This approach was compared with two existing methods to screen both genomic and metagenomic datasets obtained from the Human Microbiome Project (HMP).

**Results**. Of the methods evaluated, the new profile HMM identified the greatest number of putative LanB proteins in the stool and oral metagenome data while BlastP identified the fewest. In addition, the model identified more LanB proteins than a pre-existing Pfam lanthionine dehydratase model. Searching the gastrointestinal tract subset of the HMP reference genome database with the new HMM identified seven putative subclass I lantibiotic producers, including two members of the Coprobacillus genus.

**Conclusions**. These findings establish custom profile HMMs as a potentially powerful tool in the search for novel bioactive producers with the power to benefit human health, and reinforce the repertoire of apparent bacteriocin-encoding gene clusters that may have been overlooked by culture-dependent mining efforts to date.

Corresponding author
Paul D. Cotter, paul.cotter@teagasc.ie

## BACKGROUND

Bacteriocins are ribosomally synthesised peptides produced by bacteria that inhibit the growth of other bacteria. Some classes of bacteriocins are post-translationally modified

to provide structures beyond those possible by ribosomal translation alone. These modifications are typically key to the peptide's functionality, stability and target recognition (*Arnison et al., 2013*). Class I bacteriocins, also known as lantibiotics, are one such class of small (<5 kDa) modified bacteriocins, possessing the characteristic thioester amino acids lanthionine or methyllanthionine (*Perez, Zendo & Sonomoto, 2014*). Lantibiotics form a subgroup within the larger lantipeptide family, which also includes peptides that lack antimicrobial activity. Lantipeptides can be divided into four different subclasses (I–IV) based on the distinct biosynthetic enzymes responsible for their posttranslational modification (*Arnison et al., 2013*).

The most commonly studied lantibiotic, Nisin, is a subclass I lantibiotic, meaning that the linear prepeptide is processed by a LanBC modification system (*Arnison et al., 2013*). The core peptide undergoes a two-step posttranslational modification catalysed by two distinct enzymes—the dehydratase LanB and the cyclase LanC (*Xie & Van der Donk, 2004*). The leader sequence, necessary for recognition by the modification enzymes in the two previous steps, is then removed by the protease LanP to produce the active lantibiotic (*Xie & Van der Donk, 2004*). The gene-encoded nature of bacteriocins and bacteriocin-like peptides makes them ideal candidates for genome mining. In the case of modified bacteriocins, the structural prepeptide coding sequence often appears alongside the genes encoding proteins responsible for its modification and export from the cell. However, as more bacteriocins are discovered, the heterogeneous nature of these prepeptides is becoming ever more apparent. This diversity, coupled with their small sequence length, makes bacteriocin prepeptides much more difficult to detect using sequence-homology based searches like BLAST (*Altschul et al., 1990*). In an effort to address these obstacles, shifting the focus to the detection of bacteriocin-associated proteins opens up more avenues of discovery than simply searching for prepeptide homologs. This provides opportunities to better determine the frequency with which specific types of bacteriocin gene clusters can be found in different environmental niches, such as the human microbiota, through the investigation of metagenomic data.

It has been estimated that the human microbiota comprises approximately 100 trillion bacterial cells, outnumbering our own cells by a factor of 10 or more (*Bäckhed et al., 2005*). A recent publication, however, has argued that the ratio is actually more likely to be one-to-one, with the numbers being similar enough that each defecation event may alter the ratio to favour human cells over bacteria (*Sender, Fuchs & Milo, 2016*). Of greater consequence than bacterial numbers, however, is the collection of genes encoded in this metagenome, thought to be approximately 150 times larger than that of the human genome, with a functional potential far broader than that of its host (*Qin et al., 2010*). Regardless of absolute numbers, this dynamic community is thought to contain 100–1,000 phylotypes (*Faith et al., 2013*; *Qin et al., 2010*) and play an integral role in human health and disease (*Clemente Jose et al., 2012*; *Flint et al., 2012*). The human microbiota exhibits robust temporal stability (*Belstrøm et al., 2016*; *Jeffery, Lynch & O'Toole, 2016*) perhaps due, in part, to the protection against invading bacteria conferred by bacteriocins and other antimicrobials produced *in situ* (*Corr et al., 2007*; *Moroni et al., 2006*; *Rea et al., 2011a*). As such, investigation of the density and diversity of bacteriocins produced in the

microbiome of healthy individuals may shed light on beneficial and harmful members of this community, and key organisms for maintaining typical, i.e., health-associated, microbiota composition.

Mining the human microbiota, especially for antimicrobial compounds, has become a popular area of research in recent years (*Donia Mohamed et al., 2014*; *Walsh et al., 2015*). Due to the availability of metagenomic data generated by large public funding initiatives such as the Human Microbiome Project in the US (*The Human Microbiome Project Consortium, 2012*) and the European MetaHIT consortium (*Dusko Ehrlich, 2010*), *in silico* mining of data has emerged as a new tool that has the potential to identify antimicrobial-producing probiotics that can modulate the gut microbiota (*Erejuwa, Sulaiman & Wahab, 2014*; *Walsh et al., 2014*), or address the increasingly serious threat to public health caused by antimicrobial resistance. There are many available tools for mining the microbiome for antimicrobials, including BAGEL3 (*Van Heel et al., 2013*), antiSMASH (*Weber et al., 2015*), and traditional sequence-based approaches like BLAST (*Altschul et al., 1990*). A feature commonly integrated into these tools are Hidden Markov Models (HMM) (*Morton et al., 2015*; *Van Heel et al., 2013*; *Weber et al., 2015*), a statistical method often used to model biological data such as speech recognition, disease interaction and changes in gene expression in cancer (*Gales & Young, 2007*; *Seifert et al., 2014*; *Sherlock et al., 2013*). Profile HMMs, a specific subset of HMMs, represent the patterns, motifs and other properties of a multiple sequence alignment by applying a statistical model to estimate the true frequency of a nucleotide or amino acid at a given position in the alignment from its observed frequency (*Yoon, 2009*). Profile HMMs differ from general HMMs as they move strictly from left to right along the alignment and do not contain any cycles, a feature that makes them suitable for modelling nucleotide and protein sequence data, and have been notably utilized to detect viral protein sequences in metagenomic sequence data (*Skewes-Cox et al., 2014*). For each column in the multiple sequence alignment, the profile uses one of three types of hidden state—a match state, an insert state, or a delete state, to describe residue frequencies, insertions, and deletions, respectively (*Yoon, 2009*). Profile HMMs are potentially more sensitive than sequence homology approaches for identifying more distantly related proteins as they focus on function-dependent conserved motifs that are theoretically slower-evolving, as opposed to focusing on overall sequence similarity. Indeed, profile HMMs are known to typically outperform pairwise sequence comparison methods (such as BLAST) in the detection of distant homologs (*Park et al., 1998*), at the cost of greater computational requirements—particularly in alignment scoring and *E*-value calculation (*Madera & Gough, 2002*). Correspondingly, speed is the main advantage of BLAST over profile HMMs; however, as it is a heuristic algorithm it does not guarantee identification of the optimal alignment between query and database sequences.

In this study we designed, validated and implemented a Profile HMM to search for putative subclass I lantibiotic gene clusters in the HMP metagenomes and compared its performance to some of the tools mentioned above.

## METHODS

### Data collection

HMASM (HMP Illumina WGS Assemblies) and HMRGD (HMP Reference Genomes Data) were downloaded from the Data Analysis and Coordination Centre for the HMP (http://hmpdacc.org/). A total of 835 bacterial RefSeq protein sequences annotated as "lantibiotic dehydratase" were downloaded from NCBI Protein website (13 Apr 2015) in FASTA format (listed in Table S1).

### Building and validating the new profile hidden Markov model

A global multiple sequence alignment was generated in the aligned-FASTA format using MUSCLE (v3.8.31) (*Edgar, 2004*), and a profile HMM was built from the MSA aligned-FASTA file using the HMMER tool hmmbuild (v3.1b1 May 2013) (http://hmmer.janelia.org/). For comparison of the new model's performance, HMMER3's hmmsearch tool was used, with default parameters, to search the Pfam lantibiotic dehydratase model PF04738 against the same stool and oral HMASM assemblies (the sequences used to build this model are listed in Table S2). Positive and negative controls (listed in Table 1) were used to evaluate the model's ability to (1) accurately identify LanB protein sequences, and (2) distinguish LanB protein sequences from other, related, lantibiotic modification proteins (i.e., LanM, LanKC, and LanL). The controls were also screened using the PF04738 model, the web-based bacteriocin genome mining tool BAGEL3 (*Van Heel et al., 2013*), and a traditional BlastP using the nisin-associated lanthionine dehydratase, NisB, as the driver sequence (GenBank accession number CAA79468.1) to compare the sensitivity and specificity of each approach. A flowchart of the steps involved in building, validating and applying a profile HMM is depicted in Fig. S1.

### Target sequence translation

The HMMER3 hmmsearch tool only accepts protein sequences as targets for comparison to protein profile HMMs, so a python script was created to translate the nucleotide sequences into protein sequences. The DNA nucleotide sequences were translated in six frames using the standard genetic code.

### Metagenomic screen

The HMMER3 tool hmmsearch was used, with default parameters, to search both the new LanB profile HMM and the Pfam PF04738 profile HMM (*Punta et al., 2012*) against the stool and oral subsets of the Human Microbiome Project's whole metagenomic shotgun sequencing assemblies (HMASM). 139 stool communities and 382 communities from eight different body sites within the oral cavity were screened from the HMP database. These are listed in Table 2. As an additional comparison of performance, a traditional BlastP screen was performed on the same metagenomic samples using the previously mentioned nisin-associated lanthionine dehydratase, NisB, driver sequence. A significance cutoff of $E \leq 1 \times 10^{-5}$ was chosen for both profile HMM and BlastP methods.

### Manual examination of randomly selected gene neighbourhoods

A subset of sixty hits were randomly selected and the surrounding region examined to identify other proteins involved in lantibiotic biosynthesis. Open Reading Frames were

**Table 1** Controls used in validation of the profile HMM, listing the lantibiotic produced and the subclass of modification protein responsible for lanthionine dehydration for each strain.

| Strain | Bacteriocin | Subclass |
|---|---|---|
| *Lactococcus lactis* ssp. *lactis* S0 [a,b,c,d] | Nisin Z | LanB |
| *Lactococcus lactis* ssp. *lactis* CV56[a,b,c,d] | Nisin A | LanB |
| *Lactococcus lactis* ssp. *lactis* IO-1[a,b,c,d] | Nisin Z | LanB |
| *Bacillus subtilis* subsp. *spizizienii* ATCC 6633[a,b,c,d] | Subtilin | LanB |
| *Staphylococcus aureus* subsp. *aureus* USA300_FPR3757[a,c,d] | Bsa | LanB |
| *Streptococcus mutans* CH43[a,b,d] | Mutacin I | LanB |
| *Streptococcus mutans* UA787[a,b,d] | Mutacin III | LanB |
| *Streptococcus pyogenes* [a,b,c,d] | Streptin | LanB |
| *Staphylococcus epidermidis* [a,b,c,d] | Pep5 | LanB |
| *Lactococcus lactis* subsp. *lactis* KF147[c] | None | – |
| *Streptococcus mutans* GS-5 | Mutacin GS-5 | LanM |
| *Lactococcus lactis* subsp. *lactis* plasmid pES2 | Lacticin 481 | LanM |
| *Streptomyces cinnamoneus cinnamoneus* DSM 4005 | Cinnamycin | LanM |
| *Bacillus paralichenformis* APC 1576 | Formicin | LanM |
| *Streptococcus salivarius* plasmid pSsal-K12 | Salivaricin B | LanM |
| *Streptomyces venezuelae* ATCC 10712[d] | Venezuelin | LanL |

**Notes.**
[a] Lanthionine dehydratase protein identified by our model.
[b] Lanthionine dehydratase protein identified by PF04738 model.
[c] Lanthionine dehydratase protein identified by BlastP.
[d] Lanthionine dehydratase protein identified by BAGEL3.

**Table 2** Number of metagenomic samples per body site screened.

| Site | Number of Samples |
|---|---|
| Attached Keratinized Gingiva | 6 |
| Buccal Mucosa | 107 |
| Palatine Tonsils | 6 |
| Saliva | 3 |
| Stool | 139 |
| Subgingival Plaque | 7 |
| Supragingival Plaque | 118 |
| Throat | 7 |
| Tongue Dorsum | 128 |

identified using Glimmer v3.02 (*Delcher et al., 1999*), which were then visualised using Artemis (*Carver et al., 2012*) and blasted against the nr database using BlastP.

## Genomic screen

HMMER3's hmmsearch tool was used, with default parameters, to search the new profile HMM against the draft genomes comprising the gastrointestinal tract subset of the Human Microbiome Project's reference genome database.

## Taxonomic classification of LanB-encoding contigs

Taxonomy was assigned to LanB-encoding contigs, as assigned by our profile HMM using Kaiju (*Menzel, Ng & Krogh, 2016*). Analysis was performed in MEM run mode using default parameters and the NCBI non-redundant protein database.

## Statistical analysis

Statistical analysis was performed in R (v. 3.1.3) (*R Core Team, 2015*).

# RESULTS

## Validation of the profile hidden Markov model

The ability of the newly developed profile HMM and the Pfam lantibiotic dehydratase model PF04738 to detect LanB-encoding genes were compared using the positive and negative controls listed in Table 1. The positive controls were selected based on a relevant book chapter (*Rea et al., 2011b*) and all are previously characterised bacteriocin producers for which the sequence of the relevant biosynthetic gene cluster was available. None of the positive control sequences were used in the building of the model and a graphical representation of these clusters is presented in Fig. 1. *Lactococcus lactis* subsp. *lactis* KF147 was chosen as a negative control because it is of the same subspecies as three of the positive controls (*Lactococcus lactis* subsp. lactis S0, *Lactococcus lactis* subsp. *lactis* CV56 and *Lactococcus lactis* subsp. *lactis* IO-1) but does not produce a bacteriocin. *Streptococcus mutans* GS-5, *Streptomyces cinnamoneus cinnamoneus* DSM 4005, the *Lactococcus lactis* subsp. *lactis* IL1835 plasmid pES2, the *Streptococcus salivarious* plasmid pSsal-K12, and the newly characterised formicin producer *Bacillus paralicheniformis* APC 1576 were chosen as negative controls to evaluate the ability of the model to differentiate between LanB (subclass I) proteins and the LanM proteins-from these strains, which perform a similar, but distinct, function in the posttranslational modification of subclass II lantibiotics. *Streptomyces venezuelae* ATCC 10712 was chosen as the final negative control as it has been reported to produce a LanL-type lantipeptide (*Goto et al., 2010*). Examination of the ATCC 10712 genome using BAGEL3 identified several other orphan lantibiotic modification genes, including those encoding putative LanL, LanM, LanD and LanB proteins. The genome also appeared to encode a subclass III lantipeptide cluster comprised of genes potentially encoding a structural protein, two ABC-type transporters and a LanKC modification protein. Notably, there have been no reports of subclass I lantibiotic production by ATCC 10712 despite an in-depth investigation into the strain's lantipeptide producing capability (*Goto et al., 2010*), and BAGEL3 identified no other lantibiotic-related genes in the area of interest leading us to determine that this was a false positive.

The newly developed LanB profile HMM correctly identified the LanB protein in all nine positive controls, while the PF04738 profile HMM correctly identified the LanB protein in eight of the nine positive controls, failing to detect the Bsa-associated LanB protein in *Staphylococcus aureus* subsp. *aureus* USA300_FPR3757. Both the LanB and PF04738 profile HMMs returned no false positives when searched against the seven negative controls used in this study.

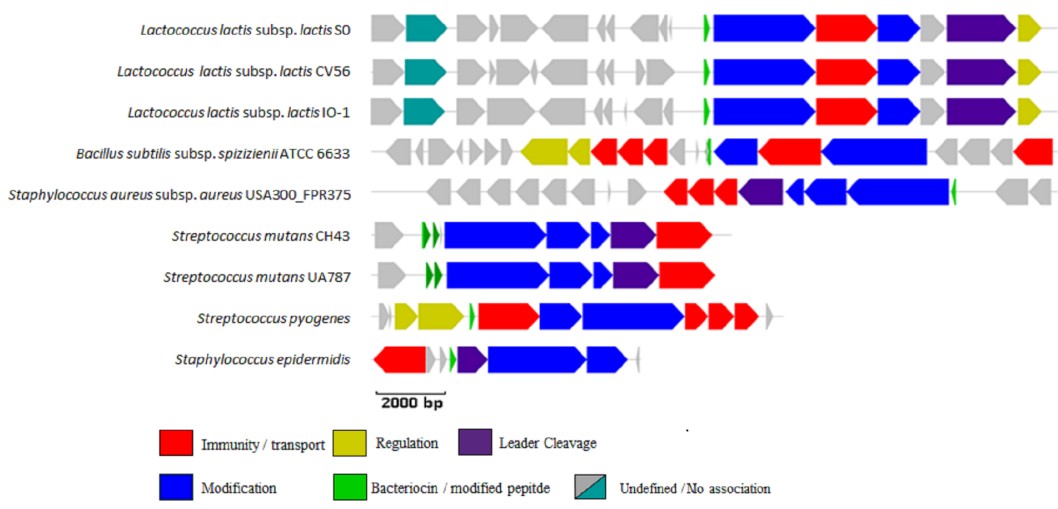

**Figure 1** **BAGEL3 output of putative bacteriocin gene clusters identified in the positive controls used for validation of our new profile HMM.** Each predicted open reading frame is colour-coded based on the role it plays in lantibiotic biosynthesis.

The web version of BAGEL3 correctly identified the lantibiotic modification proteins in all positive and negative controls, excepting the aforementioned ATCC 10712-encoded LanB concluded to be a false positive. Interestingly, examination of these controls with the BlastP method described previously, failed to correctly identify the LanB proteins encoded by *Streptococcus mutans* CH43 and *Streptococcus mutans* UA787, although the former ($E = 3 \times 10^{-4}$) fell just short of the significance cutoff ($E \leq 1 \times 10^{-5}$). BlastP also incorrectly identified a LanB protein in the negative control *Lactococcus lactis* subsp. *lactis* KF147.

## Metagenomic screen

A search with the newly developed profile HMM against the HMASM database identified 399 hits from the stool metagenomes and 1169 hits from the oral metagenomes. In contrast, the PF04738 model identified 288 hits from the stool metagenomes and 686 from the oral metagenomes. Our model reported at least one putative lantibiotic gene cluster in 81% of oral metagenomes and 86% of stool metagenomes, compared to 73% and 76%, respectively, identified by the Pfam model. The distribution of hits per sample is presented in Fig. 2. BlastP identified 231 hits from the stool metagenomes and 374 hits from the oral metagenomes. The results of these three approaches were compared to ascertain what proportion of significant hits was common to more than one search method. The results of this comparison are summarised in Fig. 3 and show that the newly developed profile HMM identified the greatest number of lantibiotic modification genes in datasets from both body sites, while the BlastP approach identified the fewest.

The overall results of these combined screening approaches, illustrated in Fig. 4 and summarised in Table S3, show a higher number and density of hits in the oral metagenomes than in the stool metagenomes (Wilcoxon rank sum test, $p = 1.399\text{e--}05$) and they also reveal a large variation in density of hits between the different sites within the oral metagenomes. This pattern was also reflected in four of the Oral subsites, namely Saliva, Subgingival

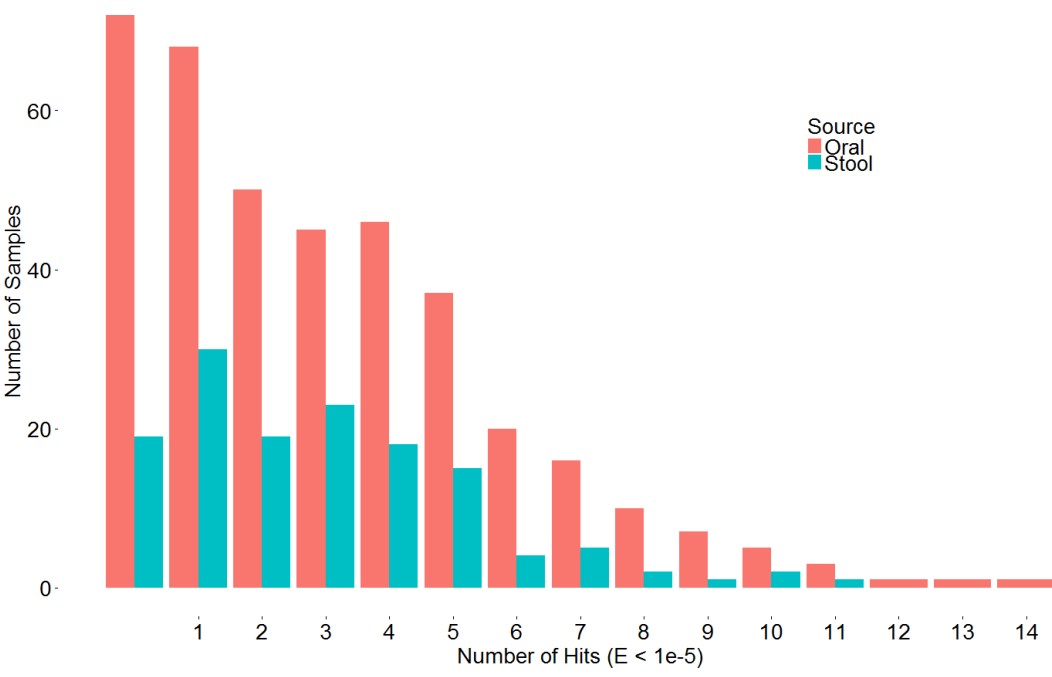

**Figure 2** Barchart depicting the distribution of lanthionine dehydratase protein numbers identified by our new profile HMM in metagenomic samples from the stool and oral microbiota.

Plaque, Supragingival Plaque and Tongue Dorsum, all of which had a significantly higher LanB density than the Stool metagenomes ($p = 0.0258, 0.0014, 6.7e–09$, and $9.4e–06$, respectively). Within the Oral samples, our model revealed a large variation in density of hits between different subsites. The throat metagenomes had the lowest LanB density, and exhibited significantly lower densities than Saliva ($p = 0.0287$), Subgingival Plaque ($p = 0.009$), Supragingival Plaque ($p = 0.0016$), and Tongue Dorsum ($p = 0.0031$) subsites'.

## Manual examination of selected gene neighbourhoods

Sixty hits, listed in Table S4, were randomly selected from those identified by the new profile HMM, 45% (27/60) of which were identified by at least one of the other two methods, and manually examined to determine if a bacteriocin gene cluster could be identified. A total of 42% (25/60) of these were not further analysed because the often relatively short regions assembled from the shotgun data prevented the identification of a full lantibiotic gene cluster. However, of the 35 remaining clusters, 28 (80%) appeared to encode multiple genes involved in the biosynthesis of bacteriocins and thiopeptides. These genes encode proteins involved in posttranslational modification, bacteriocin transport, leader cleavage and regulation (Fig. S2).

A total of 81 hits identified by BlastP were missed by both profile HMM approaches. A total of 50 of these originated in the stool metagenomes and were selected for manual annotation to determine if an overall structure or similarity could be observed. A total of 29 of these 50 were part of clusters whose components showed relatively low sequence identity

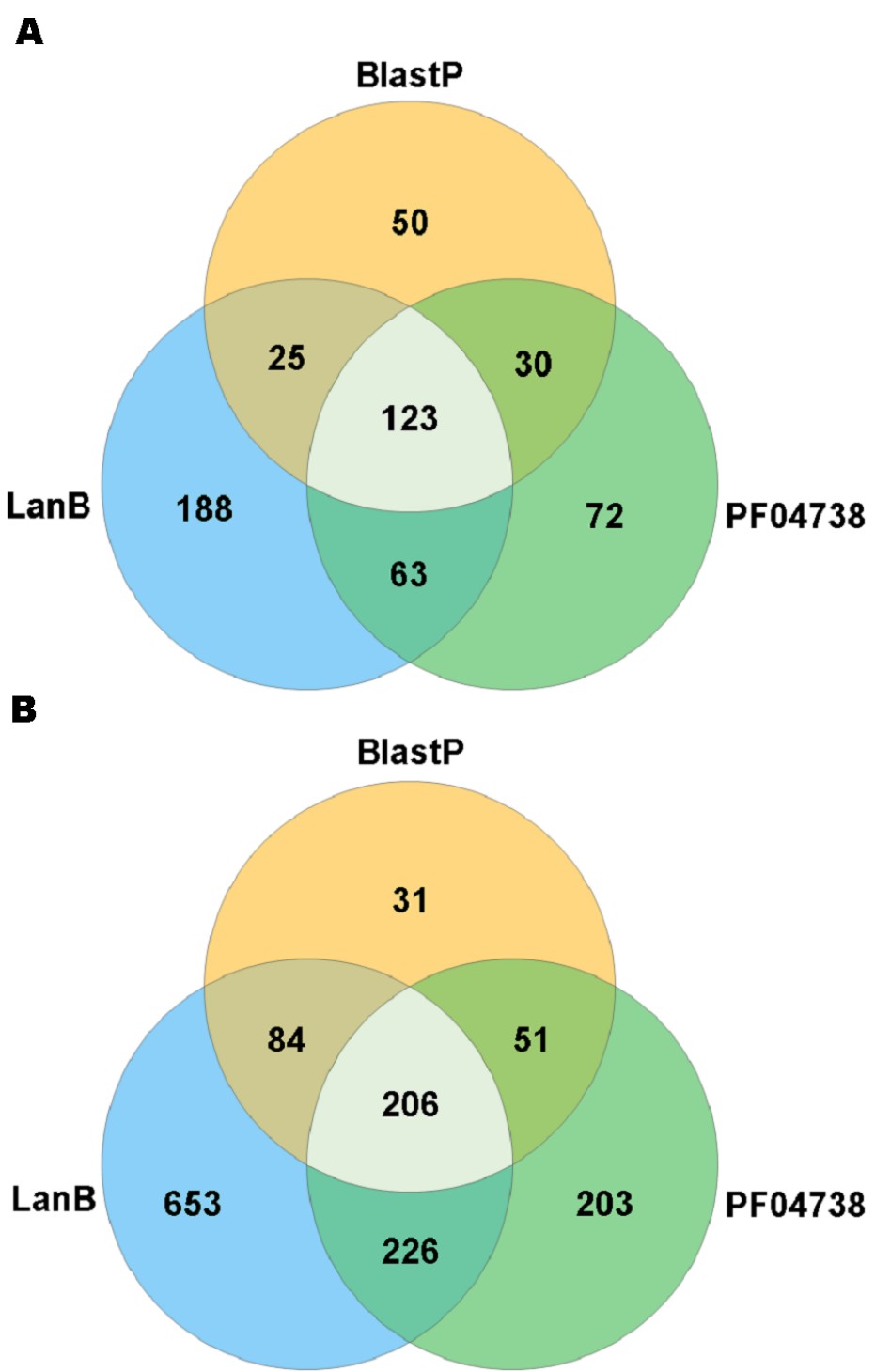

**Figure 3** Venn diagram illustrating the numbers of lanthionine dehydratase proteins reported in stool (A) and oral (B) metagenomic data by single and multiple methods.

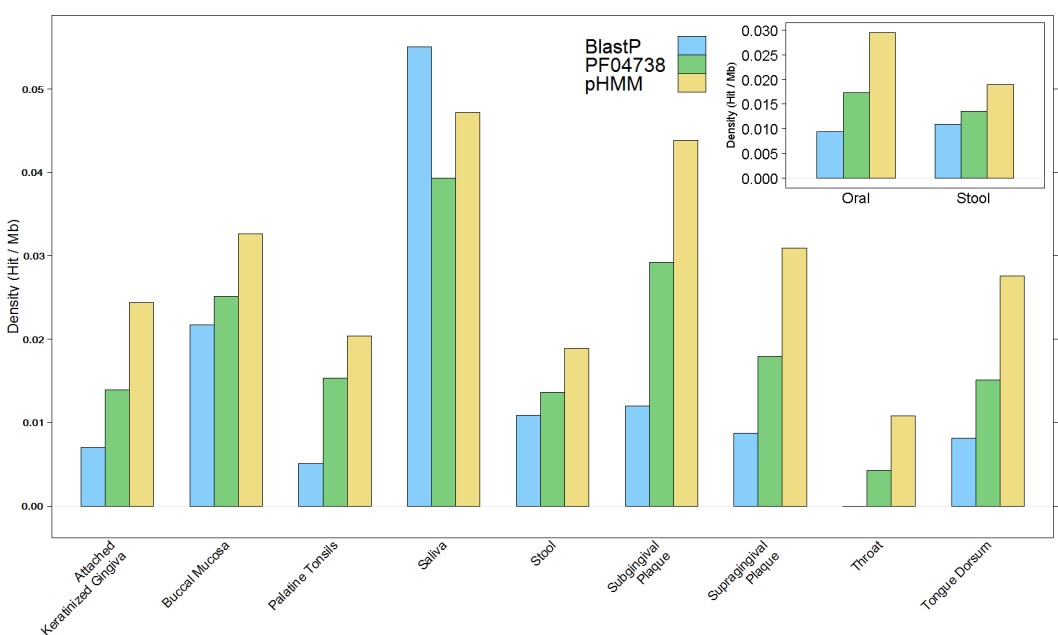

**Figure 4** **Comparison of lanthionine dehydratase density by body site reported by all three methods.** Insert shows overall comparison between stool and oral environments.

(39–50%) with proteins responsible for the biosynthesis of thiopeptides and lantibiotics, including a putative lanthionine dehydratase, a radical SAM/SPASM domain-containing protein, a thiopeptide-type bacteriocin biosynthesis domain-containing protein, an S41 family peptidase, and a protein of unknown function (DUF4932) predicted to be a putative metalloprotease. All 50 manually annotated gene clusters are available in GENBANK format and an example of this cluster architecture is summarised in Table S5.

## Genomic screen

The draft genomes of the gastrointestinal tract subset of the HMRGD were also used as a database and searched using the new profile HMM. This resulted in the identification of seven hits, including two strains of *Coprobacillus*, a potentially probiotic genus (*Stein et al., 2013*; *Yan et al., 2012*) (Table 3). From these seven genomes, only three lantibiotic gene clusters were identified by BAGEL3, these are illustrated in Fig. 5. Although this low frequency of lanthionine dehydratase proteins in the genomic dataset (0.006 hits/Mb) contrasts with the findings of the metagenome screen reported above, it is in agreement with previous reports of relatively low subclass I lantibiotic density within the human microbiota (*Walsh et al., 2015*; *Zheng et al., 2014*). A possible explanation for this significantly lower gene density (Welch's two sample $t$-test, $p = 1.232e - 10$) is that the subclass I lantibiotic clusters identified in the metagenomic data by the new profile HMM are present in the genomes of rarer members of the gut microbiota, which are not represented in the HMP reference genome database.

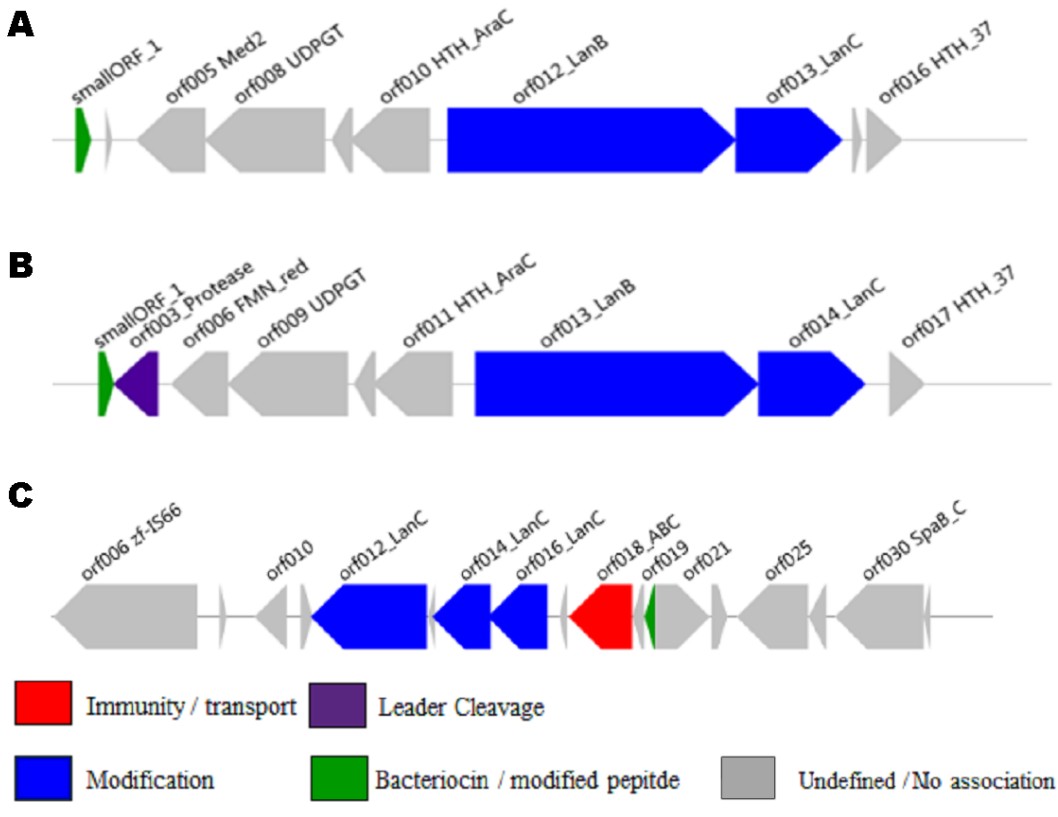

**Figure 5** BAGEL3 output of three putative bacteriocin gene clusters identified from the gastrointestinal tract subset of the Human Microbiome Project's reference genome database by our new profile HMM. (A) *Coprobacillus* sp. D6 (B) *Coprobacillus* sp. 29_1 (C) *Dorea formicigenerans* 4_6_53AFAA. Each predicted open reading frame is colour-coded based on the role it plays in lantibiotic biosynthesis.

**Table 3** Detailed information of all lanthionine dehydratase proteins identified in the gastrointestinal tract subset of the Human Microbiome Project's reference genome database using our profile HMM.

| Accession | Strain | E Value |
|---|---|---|
| JH414709 | *Bacillus* sp. 7_6_55CFAA_CT2 | 9.0E−16 |
| GL636578 | *Coprobacillus* sp. 29_1 | 3.7E−67 |
| AKCB01000002 | *Coprobacillus* sp. D6 | 4.5E−68 |
| JH126516 | *Dorea formicigenerans* 4_6_53AFAA | 2.3E−81 |
| ACEP01000029 | *Eubacterium hallii* DSM3353 | 9.4E−27 |
| KI391961 | *Fusobacterium nucleatum* subsp. *animalis* 3_1_33 | 2.2E−09 |
| GG657999 | *Fusobacterium* sp. 4_1_13 | 7.1E−09 |

## Taxonomic classification of LanB-encoding contigs

The MEM run mode of Kaiju works by searching for exact matches of given length between the query and database sequences, in the case of multiple hits of the same length in different taxa, a lowest common ancestor is inferred. Kaiju classified 378 of 399 LanB-encoding contigs. Of these, 232 were classified to the species level—however, 68 were removed as their exact species was ambiguous. Of the remaining 164 classified contigs, 66 (40.2%) were

represented at the species-level in the previously screened HMRGD database. The most abundant genus was *Alistipes,* accounting for 14.03% of LanB-encoding cotigs identified by our model, followed by *Blautia* (7.77%), *Clostridium* (4.51%), and *Bacteroides* (3.76%) (Table S6).

## DISCUSSION

Bacteriocin production enhances the competitiveness of bacteria living in complex communities and has the potential to be harnessed for the benefit of human health. The goal of this study was to develop a profile HMM and to assess its ability, in comparison with several other approaches, to detect putative subclass I lantibiotic gene clusters in human metagenomic datasets. Through this process it was also possible to evaluate the potential frequency and distribution of these bacteriocin gene clusters in the human microbiota.

To validate the model, nine positive controls and five negative controls were selected to evaluate its sensitivity and specificity. These controls were selected based on reported bacteriocin production; all positive controls were known producers of subclass I lantibiotics while the negative controls produced either different subclasses of lantibiotics or none at all. Following validation, genomic and metagenomic data corresponding to two niches within the human microbiome were chosen as the focus of this study. The first of these niches was human stool and was selected as the corresponding samples were most likely to yield bacteriocin producers with the potential to modulate undesirable microbiota profiles associated with obesity, colorectal cancer, type 2 diabetes or inflammatory bowel diseases due to their ability to survive and colonise this environment. Secondly, human oral communities were examined as a previous study showed that they contained, by far, the greatest proportion of bacteriocin structural genes across a number of human metagenome samples (*Zheng et al., 2014*). Zheng et al. reported that 80% of class I bacteriocins (lantibiotics) and 89% of all bacteriocins identified using their method originated in the oral metagenomes, while the stool metagenomes contained just 15% and 7%, respectively. The same study reported that 88% of samples from the oral cavity and 73% of samples from the gut contained at least one bacteriocin (regardless of class), while the new profile HMM reported these statistics as 81% and 83%, respectively for subclass I lantibiotics alone. The *in silico* screen carried out with the profile HMM is consistent with the observation by *Zheng et al. (2014)* by yielding a higher number and density of hits from the oral, compared to the stool, metagenomic data. Furthermore, the large variation in density of hits between sites within the oral environment suggests that lantibiotic production confers a greater advantage in saliva, subgingival plaque, supragingival plaque, and tongue dorsum communities compared to communities from the throat. This may be due to the direct benefits of antimicrobial activity but could also involve the intra- and interspecies signalling roles attributed to lantibiotic peptides (*Upton et al., 2001*), particularly in the intensely competitive microbial biofilm environment of dental plaque.

One of the most interesting observations from the study was the large variation in the numbers of *lanB* genes reported by the three different approaches. The BlastP approach identified, by far, the lowest number of significant hits overall and the lowest in every

body site examined, except for the saliva microbiome. Our model identified more than double the number of hits provided by the BlastP-based approach, in line with the aforementioned knowledge that profile HMMs can detect as much as three times as many distant homologs than pairwise methods (*Park et al., 1998*). Our model also identified a greater number of LanB proteins than the Pfam PF04738 model when used to search the same data using the same parameters. While the PF04738 model relates to the C-terminus of the lanthionine dehydratase protein, responsible for the glutamate elimination step of lantibiotic modification (*Ortega et al., 2015*), the newly developed profile HMM takes the full length of the LanB protein into consideration, thereby providing greater predictive power. Our model, in addition to identifying more potential LanB proteins, also exhibited greater sensitivity and specificity during validation than all other methods used to analyse the controls. As stated above, profile HMMs are already known to be particularly sensitive, the validation step, however, also suggests that they are more specific than the other methods evaluated as they were the only approach which did not return any false positives. When selecting the controls used to examine the performance of the different approaches, greater consideration was given to the quality of these controls than their quantity. Only controls with experimentally characterised lantibiotic production were included in the validation dataset. This relatively small control group means that, although the results of the validation step may explain the contrasting numbers of LanB proteins reported by our model and the PF04738 model, it cannot be said for certain that our model performed better.

Zheng et al. using the same metagenomic data that was the focus of this study, identified 17 potential subclass I lantibiotics from stool samples and 76 from oral samples, a much lower frequency of detection than in this study, probably due to the different methodologies used. That study focused on searching for proteins similar to those in BAGEL3's manually curated database, an approach which likely lost sensitivity because bacteriocin precursor peptides can differ considerably at primary sequence level. Furthermore, the screen employed a BLAST-based approach which, as demonstrated here, exhibited the lowest number of significant hits reported.

To investigate the areas surrounding the LanB-encoding genes identified by our model we randomly selected thirty positive hits from the oral and stool metagenome screens for manual examination. This approach revealed that several of the hits were on scaffolds that were either too small to contain a full gene or did not contain the gene's start codon. This was most likely as a consequence of the fragmented nature of the metagenomic data, as opposed to identification of true false positives by the model and would probably occur regardless of the method employed. A total of 42% (25/60) of hits selected for manual examination were discarded based on these criteria. It also revealed that a considerable number of hits exhibited low ($\sim$30%) similarity to putative thioesterases in the nr protein sequence database, highlighting that lanthionine dehydratases are relatively-closely related to proteins involved in the posttranslational modification of thiopeptides, most likely those responsible for dehydration of serine and threonine residues (*Garg, Salazar-Ocampo & van der Donk, 2013*). The similarity between these dehydratase proteins suggests a possible common ancestor protein (*Kelly, Pan & Li, 2009*). Another possible explanation relates to the fact that all of the proteins annotated as thiopeptide modification proteins are

putative annotations and none, to our knowledge, have been confirmed as such *in vitro*. It is possible, therefore, that these may simply be lanthionine dehydratases which have been incorrectly annotated due to automatic software and incomplete/under-curated databases. The majority of clusters identified contained genes encoding both LanB and LanC modification proteins, while many also contained a leader cleavage and activation peptidase and/or ABC transporter proteins for export of the mature peptide, suggesting that these have the potential to encode a functional lantibiotic.

To evaluate the model's performance in a genomic context we applied it to the gastrointestinal tract subset of the HMP's reference genome database and compared the results to our previously published study which used the online bacteriocin genome mining tool BAGEL3 to screen this same database (*Walsh et al., 2015*). The results of the two screens were startlingly different and served to highlight the variation in results that can arise from applying different methods to the same data. Interestingly, the gastrointestinal tract reference genomes encoded a significantly lower frequency of LanB hits than the stool metagenomic samples. Taxonomic classification of the 399 LanB-encoding contigs identified by our new model from the stool metagenomes revealed that only 40.2% of these potential lantibiotic producing strains were represented in the reference genome database, suggesting that the majority of these lantibiotics were encoded by rarer members of the gut microbiota or those that have not previously been identified as important. Taxonomic classification of these LanB-encoding contigs also served to highlight patterns in the results of the three approaches used (Fig. S3), for example our model identified *Allokutzneria*, *Coprococcus, Enterovibrio, Paenibacillus,* and *Tenicibaculum*-encoded LanB proteins that were completely missed by the Pfram and BlastP approaches.

## CONCLUSIONS

Across the oral and stool communities examined, this study identified 2007 unique putative subclass I lantibiotic biosynthetic gene clusters by three different methods, further emphasising the tremendous potential that the human microbiota has as a source of therapeutic compounds. As this study was performed entirely *in* silico, the next challenge lies in experimentally identifying and characterising these putative bacteriocins to identify those with the ability to desirably modulate the microbiota for the treatment of disease.

### List of abbreviations

| | |
|---|---|
| **HMASM** | Human Microbiome Project's Illumina Whole Genome Shotgun Assemblies |
| **HMM** | Hidden Markov Model |
| **HMP** | Human Microbiome Project |
| **HMRGD** | Human Microbiome Project's Reference Genome Data |

## ACKNOWLEDGEMENTS

The authors would like to thank Manimozhiyan Arumugam for helpful discussion.

### Funding

CJW, CMG and PDC are supported by a SFI PI award to PDC "Obesibiotics" (11/PI/1137). The funders had no role in study design, data collection and analysis, decision to publish, or preparation of the manuscript.

### Grant Disclosures

The following grant information was disclosed by the authors:
SFI PI: 11/PI/1137.

### Competing Interests

The authors declare there are no competing interests.

### Author Contributions

- Calum J. Walsh performed the experiments, analyzed the data, wrote the paper, prepared figures and/or tables.
- Caitriona M. Guinane conceived and designed the experiments, reviewed drafts of the paper.
- Paul W. O'Toole reviewed drafts of the paper.
- Paul D. Cotter conceived and designed the experiments, wrote the paper, prepared figures and/or tables, reviewed drafts of the paper.

### Data Availability

The sequencing data screened in this paper is available from the Human Microbiome Project's Data Analysis a Coordination Centre (http://hmpdacc.org/).

Walsh, Calum (2017): LanB HMM PeerJ. figshare.
https://doi.org/10.6084/m9.figshare.4797790.v1.

### Supplemental Information

Supplemental information for this article can be found online at http://dx.doi.org/10.7717/peerj.3254#supplemental-information.

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
