# Peer review of "A Profile Hidden Markov Model to investigate the distribution and frequency of LanB-encoding lantibiotic modification genes in the human oral and gut microbiome"

_PeerJ, doi:10.7717/peerj.3254_

## Round 0.1 · original submission · Major Revisions

As you will see, the three expert reviewers think that the work is important and needed; however, they have major concerns on how the validation & design was done—especially that only two methods were used for benchmarking/validation.

·

Basic reporting

No Comments

Experimental design

No Comments

Validity of the findings

I had a few comments included within the annotated file

Additional comments

All my comments are included within the annotated file

·

Basic reporting

The manuscript is scientifically sound and written in good English, thus conforming to PeerJ's policies with respect to form. I will suggest some improvements to the 'Background' section in order make the manuscript accessible to a more general audience, but these are simply suggestions and were not considered in my overall recommendation.

L56-86: This fragment and level of detail seems out of context and does not really provide an accessible background about the LanBC modification system.

L79-82: it's unclear what the authors mean by 'the human gene complement'.

L85-87: This sentence is speculative and gives the impression that the importance of the main research hypothesis is based on speculation. Is there any experimental research that supports or even suggests this possibility? This should be cited and explained more clearly here.

L102-112: In my understanding, the main claim of the manuscript is that hidden Markov models (HMM) are in a sense superior to homology based approaches in identifying class I lantibiotic modification genes. With this in mind, HMMs are a central part of the manuscript. I must mention then that this fragment is inaccurate in providing a concept of HMMs (L102-104) and in general does a fairly poor job in helping the non-specialized reader understand what is a HMM, how and why is it used in this kind of study. It's unclear, for example, why is it relevant for a statistical model to 'mimic the actions of ribosomes during translation'.

Experimental design

Because the manuscript is reporting a computational method, the validity of the findings are inseparable from the experimental design, I will thus comment both aspects together below.

Validity of the findings

In my opinion, the data presented by the authors is not convincing and lacks significant validation steps in order to support most of their claims.

In summary, the authors build a new database of protein sequence annotated as “lantibiotic dehydratase”, build a profile HMM based on this new database and compare the predictive power of this new model in finding LanB protein sequences, compared to a previous HMM model from Pfam, and to BlastP. The main deficiency of the manuscript is that it lacks a systematic method of validation.

In order to support the claim that such method is by any means a “powerful tool in the search for novel bioactive producers” one should at least evaluate the potential trade-off between sensitivity and specificity of the method, particularly when compared to the others. Further, because all experiments were performed in sillico (with no experimental validation) one expects that at least systematic and non-subjective approaches be used to evaluate these performances. Clearly this was not the case of the manuscript. To support this opinion I will address some specific fragments of the results:

L160-187: Basically I have three concerns:
1) How were the positive controls chosen? The manuscript mentions that it used “all previously characterized bacteriocin producers for which the sequence of the relevant biosynthetic cluster was available.” But this is far from a sufficient explanation, considering that their new method only identified one single case that was not identified by Pfams HMM. What happens if more positive controls were to be used? The claim that these were all of the available ones should be better explained and the choice should be as systematic and nonbiased as possible, since it is the only validation provided to support the use of this new HMM database.

2) The authors should consider the relevance of including Figure 2 as it seems out of context. The question is if the negative controls are in fact negative and there is nothing in this Figure that helps clarify this. This is also not clear in the text as it states that the Streptomyces venezuelae ATCC 10712 encodes LanB proteins and the lack of literature of class I lantibiotic production by this strain is not sufficient to make it a valid negative control.

3) Why wasn't the BlastP-based approach compared to this validation set?


L188-214: Because the sensitivity and specificity of the 3 methods (authors' HMM, Pfam HMM, and BlasP) is not considered, makes this result fairly unconvincing. The fact that you find more putative proteins with one method dosen't mean that you find more correct proteins (true positive) or even less false negatives and this is not addressed at all in the manuscript. For example, in the Venn diagram of Figure 4 for stool metagenomes, we see that BlastP identified 50 proteins that were not identified by any other method, while author's HMM identified 188. How many of the 50 from BlastP are correct and have been missed by the author's approach? Among the 188, how many were correctly predicted? Without systematically addressing this, the whole result and the distributions of Figure 3, could mean only that more false positives were found by this method (and HMM in general) compared to the homology-based BlastP method.

Reviewer 3 ·

Basic reporting

Regarding basic structure there are only few comments:
1) Text in the figures is often of very small font, almost not possible to read.
2) The figure legends lack on information, is not possible to understand what the figure is showing just by reading the corresponding legend.
3) In figure 1, the colors do not match the color key.
4) In figure 3, the y-axis is labelled "frequency", which doesn't tell much, does it refer to the number of samples with that number of hits below the threshold?
5) Line 66: Prepropeptide is used, whereas from that point on prepeptide is used.

Experimental design

The question regarding the presence and diversity of bacteriocins in human microbiomes is relevant and meaningful. The methods proposed for investigating the question (using pHMM) are relevant and appropriate. However I have some major concerns in this point:
1) Only 3 methods are analyzed, the proposed pHMM, another available pHMM from a public database and Blast, each method has its own advantages and limitations which are not well stated in the document.
2) The sensitivity and specificity of an pHMM depends on the sequences aligned to built it, those sequences were never mentioned in the manuscript.
3) The public pHMM from pfam also has available the sequences used to built it, those should be mentioned at least in a supplementary table. Differences in the set of sequences used for the generation of the pHMM are likely a key component for differences obtained in results.
4) In the generation of the model it is said that the multiple alignment is used, however it is not clear if the full alignment was used or only a conserved region.
5) In the results section, the program Bagel 3 is mentioned, however it is not mentioned anywhere in the methods.

Validity of the findings

The findings regarding the identification of potential bacteriocins in the microbiome is novel, however, this is only a bioinformatic prediction, experimental validation is needed to proof their presence. On the side of the method, which seem one of the main aims of the manuscript, as mentioned in the manuscript, the use of pHMM for identification of diverse proteins is not novel and the fact that it finds better results than blast is also expected.

Some concerns regarding the findings:
1) The controls consist of only 9 positive controls and 5 negative controls, since it is not known how many sequences were used to build the model is hard to address if those numbers are adequate for validating the method.
2) Is not clear either if the 5 negative controls correspond to the closest possible sequences (by sequence identity) that do not correspond to the family of proteins aimed to identify. In other words, your best negative controls will be the most similar sequences available that are not LanB, and this is not clear from the manuscript.
3) No sequences of the positive controls should have been used to train the model, but this is not clarified in the document.
4) When describing the positive and negative controls, it is mentioned the identification of bacteriocin proteins in the datasets, is not clear if that identification was done using Bagel 3 or the pHMM, in case is the later, it will be suggesting false positives.
5) It is claimed that the pHMM has better performance than the pfam pHMM, but the difference is only one more positive contig identified, I don't think the difference could be considered significant in order to be able to claim which method perform better.
6) Additionally, only having 5 negative controls leaves the reader without the certainty of the precision of the method, is easy to believe that in a larger dataset false positives could be identified.
7) When the search of hits in the metagenomes is done a number reported in terms of hits/Mb is shown, however, when analyzing the genomes only the number of hits is reported and is claimed to be lower than expected. A measurement of hits/Mb should be done too in order to have a comparable value.
8) The final argument for having a lower number of genomes containing bacteriocins than what they expected based on the metagenomes is that potentially the contigs containing LanB came from rare taxa not represented by the assembled contigs. In this point I think that at least an effort to taxonomically annotate or assign the metagenomic contigs and then a comparison with the taxonomy of the assembled genomes will give further information about the likelihood of this argument.

Additional comments

Two additional comments about the figures:
1) In Figure 1 and 2, it will be desired that the coordinates at the beginning and end of each fragment in the reference sequences should be added, will help the reader get a better idea of the fragment being analyzed. The annotation by colors of the fragments seem to suggest that the majority of the genes in those regions are of unknown function, is this the case?

2) In supplemental figure 1, if the surrounding region of the predicted bacteriocin is being analyzed, but the LanB itself is not shown in the figure, why? It will be desirable to have the sizes of the contigs? Are the full contig being analyzed in all cases?

---

## Round 0.2 · Minor Revisions

There are just a few minor issues to be addressed (as stated by 2 reviewers), but it shouldn't take much time...

·

Basic reporting

Line 168: provide some information about the MEM run mode.

Lines 208-209: E values are missing a negative sign for the power of 10.

Text in the figures 2 and 4 has small font and hard to read.

Figure 2: The authors may consider combining the 2 bar charts in one and colour stool and oral samples with different colors.

Experimental design

The model for identifying this subclass of bacteriocins is interesting and could be used for other types of bacteriocins types. For that, the authors may consider including a flowchart of building the model if it is to be applied for identifying other classes of bacteriocins.

Validity of the findings

For the conclusion part, the authors mentioned the potential usage of the putative bacteriocins identified in therapy. But, The authors are advised to mention that because this study is entirely in-silico, first we need to experimentally identify and characterize these bacterioicns and whether if they are true bcteriocins or not and their effectiveness.

·

Basic reporting

The authors have satisfactorily addressed my previous comments.

Experimental design

no comment

Validity of the findings

Although the authors have addressed most of my concerns and were very kind to explain their changes in the rebuttal letter, I am still intrigued by the conclusion that more LanB genes are found through their method of profile HMMs compared to the other methods. They validate this by choosing manually 60 random hits only from the new profile HMM (lines 238-245). Based on the Venn diagrams, If you select random reads from the profile HMM, one expects roughly 50% of the hits to be identified by other methods. From this selection, only 46% turned out to be genes involved in the biosynthesis of bacteriocins and appear to be true positives. Of these 46%, how many were found by the other methods? The authors should include a table with the details of the manual curation, since it is impossible to know from supplementary table 4 if the hit was manually curated and if it returned a true positive. Conversely, I still could not find the reason why the manual curation was not made on the BLAST hits that were not found by the profile HMM as I asked in my previous review. In the rebuttal letter, the authors claim that "Applying the BlastP approach to the positive and negative controls, as suggested by the reviewer, has shown that the BlastP approach used in this manuscript suffers from sensitivity and specificity issues which may explain the 50 proteins it identified that were not identified by any other method. " But I really wish that this would be systematically addressed by manually inspecting the reads.

Additional comments

The issues pointed above significantly impact the overall quality of the manuscript in my point of view, but seem like fairly simple issues to solve. I would also suggest in a future version to mention the E-value cutoff only once in the methods rather than throughout the results, since the same cutoff is used and is repeated 7 times in the text.

Reviewer 3 ·

Basic reporting

No comment

Experimental design

I think all major concerns were answered appropriately.

Validity of the findings

No comment

Additional comments

I think the manuscript has significantly improved.

---

## Round 0.3 · accepted · Accept

Thanks for making the final corrections.